# A light-entrained clock mechanism in a hydrozoan jellyfish synchronizes evening gamete release

Ruka Kitsui[1], Noriyo Takeda[2], Evelyn Houliston[3], Ryusaku Deguchi[1]*, Tsuyoshi Momose[3]*

1 Department of Biology, Miyagi University of Education, Sendai, Miyagi, Japan, 2 Graduate School of Integrated Sciences for Life, Hiroshima University, Hiroshima, Japan, 3 Laboratoire de Biologie du Développement (LBDV), Sorbonne Université, CNRS, UMR7009, Villefranche-sur-Mer, France

* deguchi@staff.miyakyo-u.ac.jp (RD); tsuyoshi.momose@imev-mer.fr (TM)

## Abstract

For marine species that reproduce by external fertilization, spawning is precisely coordinated within a local population to maximize the chances of producing offspring. Gamete release is often synchronized with respect to the diel light changes at dawn and dusk. In the hydrozoan jellyfish *Clytia hemisphaerica*, spawning occurs when oocyte maturation and gamete release are induced by maturation-inducing hormone (MIH) neuropeptides released from opsin-expressing cells in the gonad, directly upon light stimulus. Here, we characterize the distinct spawning cycle of a previously undescribed species *Clytia* sp. IZ-D, identified on the Pacific coast of Japan, which releases gametes in the evening. *Clytia* sp. IZ-D jellyfish spawn 14 hours after a light stimulus under a 24-hour light cycle and exhibit autonomous and synchronized spawning cycles with a 20-hour interval under constant light. We find that the female spawning cycle reflects the oocyte growth and their acquisition of competence for maturation, such that each day a new batch of growing oocytes becomes responsive to MIH at a time that correlates with the timing of actual spawning. We propose that the synchronized evening spawning in this species is controlled by an atypical circadian timing mechanism based on the progressive development of gamete competence to MIH and modulation of the opsin-controlled MIH signaling pathway. This mechanism may provide resilience to light cycle instability due to local climate variation and ensure reproductive isolation from other *Clytia* species by shifting the gamete release timing.

## Introduction

Marine organisms rely on various environmental cues to synchronize sexual behavior and gamete release, thereby maximizing the chances of successful sexual reproduction. Precise synchronization of spawning timing often depends on daily changes in

**Data availability statement:** The datasets for sea surface temperature, sunrise/sunset time, spawning timing, numbers of released oocytes, delay from MIH addition to GVBD, IPBF or IIPBF (Figs 1–7, S1 and S3) are available at https://doi.org/10.5281/zenodo.17266785.

**Funding:** This work was supported by Grant-in-Aid for Scientific Research (Grant No. 20K06736), Japan Society for the Promotion of Science (https://kaken.nii.ac.jp/ja/grant/KAKENHI-PROJECT-20K06736/) awarded to RD, Sasakawa Scientific Research Grant (2025-4059), awarded to RK and CNRS Biology's International Call "Marine Biology (2023)", awarded to EH (Project Lightswitch). The sponsors played no role in the study design, data collection and analysis, publication decision, and manuscript preparation.

**Competing interests:** The authors have declared that no competing interests exist.

**Abbreviations:** GV, germinal vesicle; GVBD, Germinal Vesicle Breakdown; hps, hours post-spawning; IPBF/IIPBF, first and second polar body formation; MRC, MIH-reactive competence; MIH, maturation-inducing hormone; MIH-R, MIH-Receptor.

sunlight luminosity, which serve as reliable external triggers for marine animals. For instance, dozens of coral species release gametes simultaneously within a few hours after sunset, a few nights after the full moon in late spring on the Great Barrier Reef. This occurs despite the risk of species hybridization [1,2], likely using sunlight as the primary external signal for synchronizing gamete release. For many marine animals, spawning control is complex, integrating multiple environmental cues, and its molecular basis remains uncharacterized. For example, annual coral gamete release is influenced by wind [3], solar irradiation [4], temperature [5], and the lunar cycle [2,6,7], in addition to the daylight cycle [2,8].

The molecular basis of light-controlled oocyte maturation and gamete release is relatively well described in hydrozoan jellyfish, where daily gamete release is often directly triggered by light stimuli [9]. In *Cytaeis uchidae* and *Clytia hemisphaerica*, exposure to light after a few hours of darkness almost immediately triggers oocyte meiotic maturation [10–12]. Eggs are released as meiotic divisions are completed, shortly after sperm release from male jellyfish gonads. Both processes are initiated by light-induced secretion of a maturation-inducing hormone (MIH) from neural-type cells in the gonad ectoderm. MIH comprises short neuropeptides (e.g., WPRPamide, RPRPamide) common between hydrozoan jellyfish *C. hemisphaerica* (Leptothecata) and *Cladonema pacificum* (Anthoathecata). These MIH-secreting cells also express opsin genes [13]. In *C. hemisphaerica*, knockout of *CheOpsin9*, which is predominantly expressed in the gonad, prevents light-triggered MIH release. Secreted MIH binds to a seven-transmembrane G-protein-coupled receptor MIH-receptor (MIH-R) on the oocyte surface, triggering oocyte maturation via Gαs, cAMP, and cAMP-dependent protein kinase signaling [12].

Spawning in some hydrozoan jellyfish populations can be triggered by light-to-dark transitions rather than dark-to-light, as seen in subspecies of *Cladonema pacificum* and *Spirocodon saltatrix* [14,15], or by both transitions in some *Clytia* populations [16]. In 'dark-type' female *Cladonema*, a 3–5-minute dark pulse promotes oocyte maturation, with spawning occurring after 35 min. These jellyfish may employ different opsins and downstream signal transduction pathways to trigger MIH release from neurons. In both light- and dark-type jellyfish, oocyte maturation is initiated by MIH release, triggered by light or dark stimuli, respectively [17].

The involvement of circadian clock mechanisms in cnidarian spawning timing regulation, alongside the mechanisms that promote immediate gamete release upon light or dark stimuli, remains to be tested. Light-controlled circadian behaviors have been described in several cnidarians. Both *Hydra* (Hydrozoa) polyps and the upside-down jellyfish *Cassiopea* (Scyphozoa) exhibit cyclic sleep-like state [18,19], while calcification in the reef coral *Acropora eurystoma* (Anthozoa) occurs during the day and maintains circadian oscillations under constant light [20]. The sea anemone *Nematostella vectensis* (Anthozoa) exhibits differential locomotive behaviors between day and night, which persist under constant darkness [21]. Furthermore, components of evolutionarily conserved circadian clock genes, *Clock* and *Cycle* (*BMAL1*), have been identified in anthozoans [22,23]. The *Clock* orthologue in *Nematostella*, *NvClk*, is required for maintaining circadian locomotor rhythms under constant

light [24]. Despite these findings, circadian clocks have not previously been implicated in the temporal control of oocyte maturation and gamete release in hydrozoans. Orthologues of *Clock* and *Cycle* (*BMAL1*) appear to be absent from the genomes of *Hydra* and *C. hemisphaerica* [25] and were likely lost in the hydrozoan common ancestor. Additionally, the transcription-translation feedback loop model underlying CLOCK-driven circadian clocks may lack sufficient precision to trigger rapid events such as oocyte maturation or gamete release, which occur within minutes in hydrozoans. Consistent to this, hydrozoan gamete release can be directly altered by manipulating light–dark cycles, provided a minimum latent interval is respected [12].

Here, we present the first evidence of an autonomous clock mechanism precisely controlling and synchronizing the timing of oocyte maturation and gamete release in the hydrozoan jellyfish *Clytia* sp. IZ-D, a close relative of the model species *C. hemisphaerica*. In natural conditions, gamete release in this species occurs shortly after sunset. Through a series of experiments with intact jellyfish and isolated oocytes, we define how light separately regulates both oocyte growth and the precise timing of spawning in *Clytia* sp. IZ-D, and propose a simple model for this autonomous and nightly gamete release.

## Results

Jellyfish of a new *Clytia* species (*Clytia* sp. IZ-D) were collected on Izushima Island (38.440501°N, 141.525718°E), Miyagi, along the Pacific coast of Japan (Fig 1A) on 12 October 2023. The water temperature was 21.3°C at 2 m depth. The average sea surface water temperature in this area in mid-October from 2014 to 2023 was 18.9°C (Fig 1B), indicating that seawater temperatures were elevated in 2023. Daylight duration was approximately 11.5 hours (Fig 1C). We established a female strain (D-A1) and a male strain (D-A3) as vegetatively grown polyp colonies from planula larvae siblings, offspring of a single male–female pair of jellyfish kept in a plastic dish. We reared clonal jellyfish liberated from these colonies for use in this study. In the same location, other species, notably *Clytia* sp. IZ-C (collected in July 2023, sea surface temperature 19.4°C) were also sampled. Taxonomic analysis using maximum likelihood phylogenetic analysis of the mitochondrial 16S rRNA gene indicated that *Clytia* sp. IZ-D was distinct from any previously barcoded *Clytia* species, including the hydrozoan model species *C. hemisphaerica* (Fig 1D). However, the morphology and life cycle of *Clytia* sp. IZ-D (Figs 1E and S3) were indistinguishable from those of *C. hemisphaerica*. The life cycle consisted of a polyp colony with multiple feeding polyps (gastrozooids) and jellyfish budded from gonozooid polyps. Budded jellyfish grow to reproductive maturity within 10 days.

*Clytia* sp. IZ-D exhibited physiological differences from *C. hemisphaerica*. Notably, this species releases gametes at night in the laboratory at the standard culture temperature of 21°C, whereas *C. hemisphaerica* releases eggs and sperm within 100–120 min after light exposure following 3 hours of darkness [26].

### Sunrise triggers *Clytia* sp IZ-D ovulation on the following evening

To investigate how gamete release is controlled in *Clytia* sp. IZ-D jellyfish, we primarily focused on ovulation (egg spawning) in females using strain D-A1. Jellyfish were maintained under standard 12-hour light/12-hour dark (L12/D12) cycles at 21°C. Under these conditions, eggs containing a clearly visible female pronucleus, which indicates completion of meiosis, were released from the gonad approximately 2 hours after the light-to-dark transition (Fig 2A and 2B and S1 Movie). We defined the gamete release time of an individual jellyfish as the time point when the first egg or sperm release was observed.

Initially, we hypothesized that gamete release in *Clytia* sp. IZ-D was induced by the onset of darkness, as reported for a night-spawning subspecies of *Cladonema pacificum* [12] and certain *Clytia* species from the North Pacific [16]. Contrary to our expectations, altering the onset of the dark period (dusk) did not affect the ovulation timing (Fig 2C). However, advancing the start of the light period (dawn) by 2 hours, to 8 hours post-spawning (hps), led to ovulation occurring two hours earlier. In all tested cases in Fig 2C, ovulation occurred 14 hours after the preceding dark-to-light transition, even when

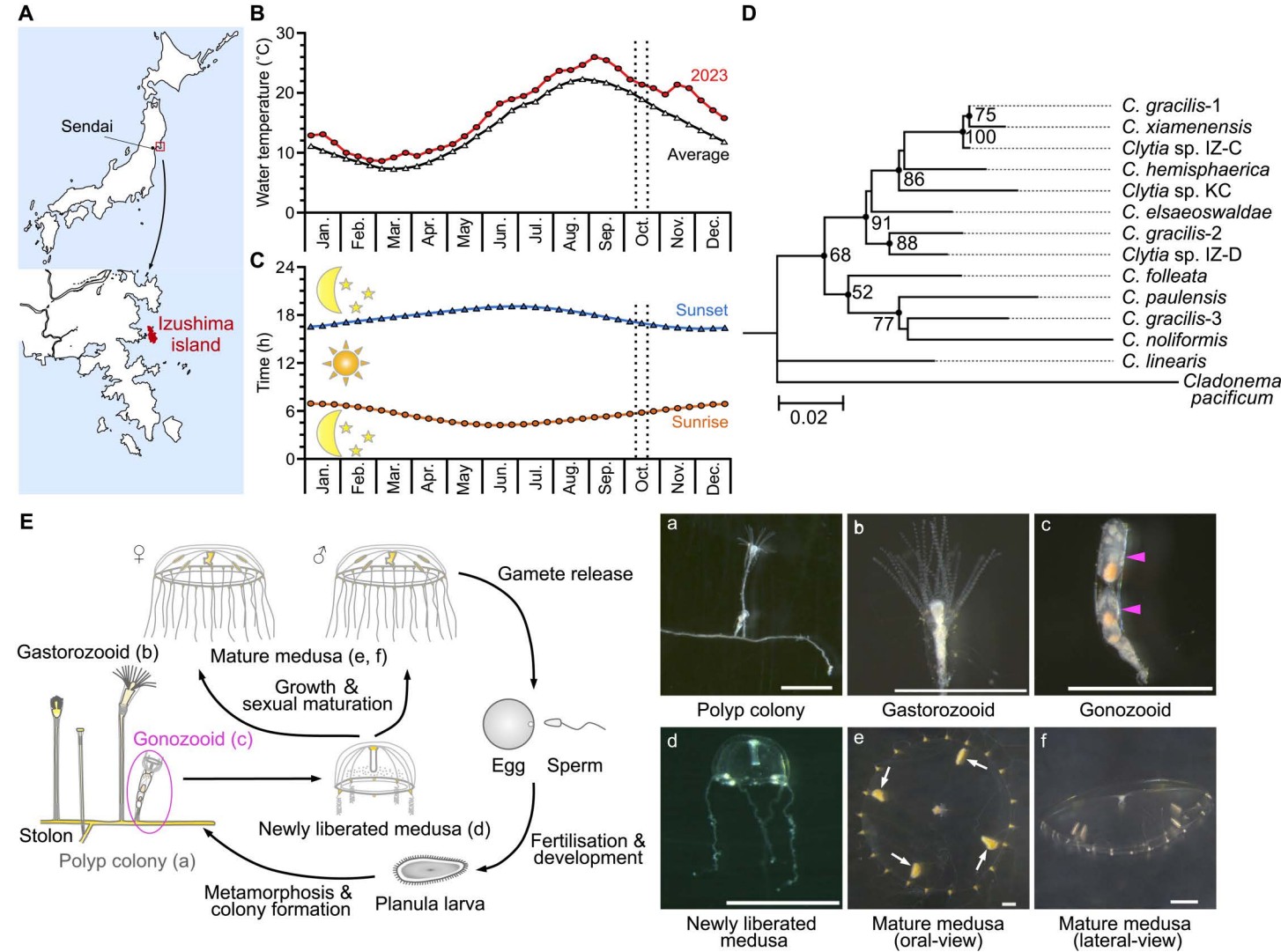

**Fig 1. _Clytia_ sp. IZ-D from Izushima Island, on the Pacific coast of Japan. (A)** Geographical location of the sampling. Source: Geospatial Information Authority of Japan (https://maps.gsi.go.jp/vector/) with PDL 1.0 (https://www.digital.go.jp/en/resources/open_data/public_data_license_v1.0). **(B)** Annual sea surface temperatures in 2023 (sampled year) and 10-year average, observed at Enoshima Island, 8 km southeast of Izushima. **(C)** Annual change of the sunrise and sunset times. The sampling period (mid-October in 2023) is indicated by vertical dotted lines in B and C. **(D)** Molecular phylogeny of 16S rRNA sequence from _Clytia_ species obtained by Maximum-Likelihood analysis. Bootstrap values of 1,000 pseudoreplicates above 50% were shown as node-support values. **(E)** Life-cycle diagram (left) and images of each stage (a–f) of _Clytia_ sp. IZ-D, which is morphologically very similar to the model species _Clytia hemisphaerica_. Magenta arrows in the gonozoid cartoon (c) indicate individual medusa buds. Arrows in the mature female medusa photo (e) point to the four gonads in the subumbrella. Meteorological data subsets used in Fig 1B and 1C are available at https://doi.org/10.1101/2025.05.05.651927.

the light period was extremely short (as brief as one hour or one minute; Fig 2D). These observations strongly suggest that the dark-to-light transition, which we hereafter refer to as "light stimulus", acts as a trigger for ovulation with a 14-hour delay. Such a delayed response has not previously been described. When jellyfish were subjected to 24-hour cycles with different daytime lengths (8 and 16 hours of light), ovulation consistently occurred 14 hours after the light stimulus, thereby resulting in the 24-hour cycle (Fig 2E). Under L16/D8 cycles (16 hours of light and 8 hours of dark), ovulation occurred during the light phase (2 hours before sunset), again confirming that the light stimulus is responsible for inducing

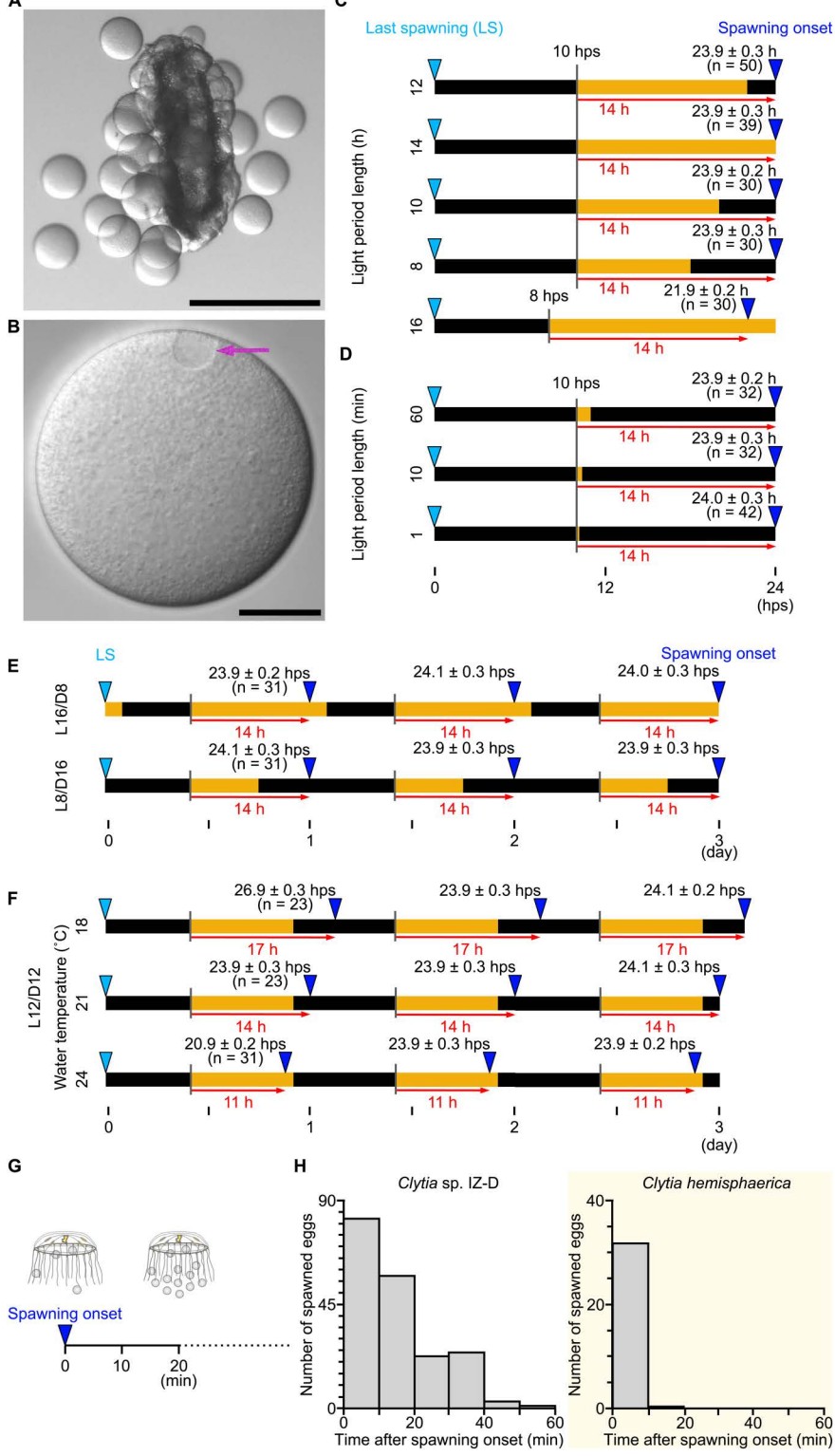

**Fig 2. Ovulation occurs 14 hours after a light stimulus at 21°C. (A)** Snapshot image of eggs being released from an isolated female gonad, which occurs with the same timing as in intact jellyfish. Scale bar: 0.5 mm **(B)** A mature oocyte released from the gonad. Magenta arrow indicates the female pronucleus. Scale bar: 50 μm **(C)** The timing of spawning onset when the start or end of the light period was changed; spawning occurs 14 hours from

the onset of light. **(D)** A short light period was sufficient to induce spawning 14 hours after light onset. **(E)** Daily regular spawning of jellyfish maintained under a constant light–dark cycle; spawning started about 14 hours after each light stimulus, occurring either before or after the end of the light period, depending on its length. **(F)** Temperature dependence of the interval between light stimuli and spawning. Jellyfish spawned regularly 17, 14, and 11 hours after the light stimulus at 18°C, 21°C, and 24°C, respectively, under an L12/D12 cycle. For all experiments shown in (C–F), jellyfish were maintained in L12/D12 at 21°C until the light–dark cycle or temperature was changed just after the last spawning (LS: indicated with light blue triangles). Horizontal bars represent the light and dark periods, with LS as the origin of the time scale. Key intervals from the light stimuli to the spawning, likely constraining the spawning schedule, are annotated with red arrows with approximate lengths in hours; $n$ = number of jellyfish used for each condition. **(G)** Experimental design for individual oocyte release timing measurement. The number of eggs released from a jellyfish was counted every 10 min following the first egg release. **(H)** Timing of individual oocyte release from the onset of ovulation for each jellyfish, comparing Clytia sp. IZ-D ($N$ = 10) and *Clytia hemisphaerica* ($N$ = 10). Spawning observation data are available at https://doi.org/10.1101/2025.05.05.651927.

ovulation. The onset of ovulation for most jellyfish varied by only ± 0.3 hours among individuals cultured under the same conditions (Fig 2C–2E). This synchronization is notable given the 14-hour delay between the light stimulus and ovulation. To explore the precision of this delay and the factors influencing it, we examined conditions involving switches to higher (24°C) or lower (18°C) incubation temperatures (Fig 2F) while maintaining the standard L12/D12 light cycle. Ovulation occurred 11 or 17 hours after the light stimulus at 24°C and 18°C, respectively, compared to 14 hours at 21°C. Despite this temperature-dependence, the variance in ovulation timing within each cohort remained low (±0.2 − 0.3 hours). We conclude that the light stimulus, corresponding to sunrise in the natural habitat, robustly determines the well-synchronized timing of evening ovulation. Similarly, male jellyfish (D-A3) released sperm 13 hours after the light stimulus (S1 Fig and S2 Movie) at 21°C.

These results collectively indicate that *Clytia* sp. IZ-D possesses an unusual "delayed response" timer mechanism for synchronizing evening gamete release. We further noticed that egg release from *Clytia* sp. IZ-D jellyfish continued for up to 60 min from its onset, compared to 10 min for the morning-spawning *C. hemisphaerica* (Fig 2G and 2H). This difference suggests that in *Clytia* sp. IZ-D, the timing of oocyte maturation induction may be partly controlled independently at the level of individual oocytes, whereas in *C. hemisphaerica*, MIH release is the dominant factor [13].

### *Clytia* sp. IZ-D ovulation is regulated by a light-entrained clock

Ovulation induced directly by a light-to-dark transition is a common strategy for species that release gametes in the evening [14,15]. Based on the observations presented above, we speculated that the extended delay from the light stimulus to ovulation observed in *Clytia* sp. IZ-D reflects a distinct circadian mechanism regulating gamete release, involving factors that operate autonomously within individual oocytes. To test this hypothesis, we placed female jellyfish under constant light. Notably, *Clytia* sp. IZ-D spawned eggs every 19–20 hours over at least three ovulation cycles at 21°C (Fig 3A). The interval of ovulation was temperature-dependent, occurring every 17–19 hours and 20–22 hours at 24°C and 18°C, respectively (Fig 3A), matching the temperature-dependency of the delay of ovulation from the light stimuli. Across all temperature conditions, the variance in ovulation onset was greater (up to 0.9 hours) under constant light than under a light–dark cycle, particularly during the second or third ovulation cycle. This increased variance may partly reflect the accumulation of variation over successive ovulation cycles, however, we observed that egg release from a single jellyfish continued longer, up to 80 min, under constant light (Fig 3B). This supports the hypothesis that the autonomous ovulation cycle is controlled at the level of individual oocytes. Altogether, we conclude that *Clytia* sp. IZ-D has autonomous gamete release cycles operating at around 20 hrs per cycle under constant light. We term it a 'quasi-circadian' cycle because the cycle duration is shorter than 24 hours and highly temperature-dependent, unlike authentic circadian oscillators [27,28]. A reliable daily dark-light cue will entrain this quasi-circadian ovulation rhythm to a 24-hour cycle, synchronizing gamete release among jellyfish under the same environmental conditions and among oocytes within an individual jellyfish.

The observation that ovulation induced by light stimuli occurs later than for jellyfish placed under constant light seemed curious if the dark-to-light stimulus were indeed a positive trigger for ovulation. To examine how the light stimulus entrains

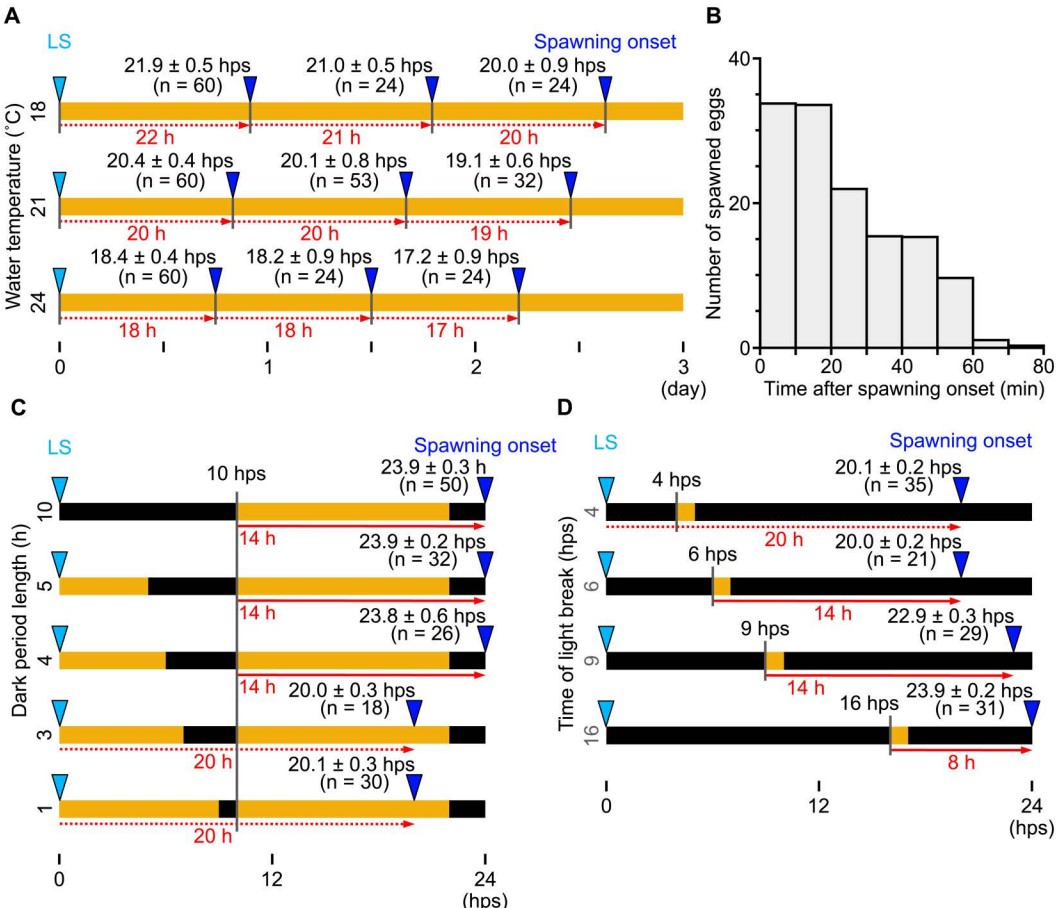

**Fig 3. Autonomous synchronous ovulation cycles under constant light in *Clytia* sp. IZ-D.** Spawning behavior was monitored following different light-dark regimes, represented graphically as in Fig 2. **(A)** Cycles of synchronous egg spawning over three days under constant light. **(B)** Individual egg release from jellyfish (*N* = 10) under constant light, measured from the onset of spawning for each jellyfish. **(C)** Demonstration of a 4-hour minimum dark period required before the dark-to-light transition, so that it modulates the spawning as a light stimulus. **(D)** Influence of light stimulus timing on spawning. Spawning occurred at 20 hps when the light stimulus was applied 6 hps or earlier, and not later than 24 hps when applied 10 hps or later. Egg spawning is thus constrained to a window between 20 and 24 hps. In **(A)**, **(C)**, and **(D)**, jellyfish were maintained in the standard light–dark cycle of L12/D12 at 21°C and shifted to the experimental conditions after the last spawning (LS). Red dotted arrows and numbers indicate the interval between each spawning. Red arrows and numbers represent the key approximate intervals between the light stimulus and the spawning, when the spawning cycle differs from the autonomous spawning cycle (i.e., 20 hps at 21°C). Spawning observation data are available at https://doi.org/10.1101/2025.05.05.651927.

the autonomous ovulation cycle, we altered the duration of the dark period while maintaining the timing of the dark-to-light transition after the previous ovulation (at 10 hps; Fig 3C). As described above, the dark-to-light shift induced (delayed) ovulation after 14 hours when the dark period was four hours or longer (Fig 3C). In contrast, ovulation occurred at 20 hps when the dark period was three hours or shorter, matching the timing observed under constant light. This indicates that a dark period of at least four hours is necessary for the dark-to-light transition to act as an effective light stimulus. Similarly, minimum dark periods are required for jellyfish that spawn eggs immediately after the light stimulus, including *C. hemisphaerica* [12,26], probably to allow recovery or re-sensitization of opsins to light.

We then addressed how the light stimulus interacts with the 20-hour autonomous ovulation cycle. For instance, the action of the light stimulus in *Clytia* sp. IZ-D could be explained either by a delaying effect on the autonomous cycle or by resetting or overriding the cycle to initiate a new 14-hour timer. To address this, we substantially altered the timing of the

light stimulus, following a preceding dark period of four hours or more (Fig 3D). A light stimulus earlier than 6 hps induced ovulation at 20 hps, equivalent to the autonomous ovulation cycle under constant light. Light stimuli between 6 and 10 hps induced ovulation 14 hours later, as characterized under the standard light cycle (Fig 2A). A third situation was observed when the light stimulus was applied 10 hps or later, with ovulation occurring at 24 hps, less than 14 hours after the light stimulus (Fig 3D). This last observation is inconsistent with a model in which light stimulus resets the autonomous ovulation cycle and starts a new 14-hour timer. The effect of the light stimulus is thus better described as delaying the autonomous 20-hour ovulation cycle, acting within a light-sensitive window between 6 and 10 hps.

## Asynchronous ovulation under constant darkness suggests a permissive role of light to maintain the circadian clock

The results presented above collectively indicate that daily light cues entrain the autonomous quasi-circadian gamete release cycle. We found that light also plays a permissive role in maintaining the cycle, such that jellyfish release eggs asynchronously under constant darkness. To do this, we transferred jellyfish from constant-dark conditions after 20, 24, or 28 hps and counted eggs already released, or released within one hour. (Fig 4A). At 20 hps, the time of autonomously spawning under constant light conditions, no eggs had yet been released, and 23% of jellyfish spawned within the hour following exposure to light. At 24 hps, 20% of jellyfish maintained in the dark had spawned, with a further 39% spawning within the following hour. By 28 hps, most jellyfish (74%) kept in darkness had completed spawning, with the remainder spawning within the hour (Fig 4B). In these experiments, although most oocytes were released within 1 hour of the first

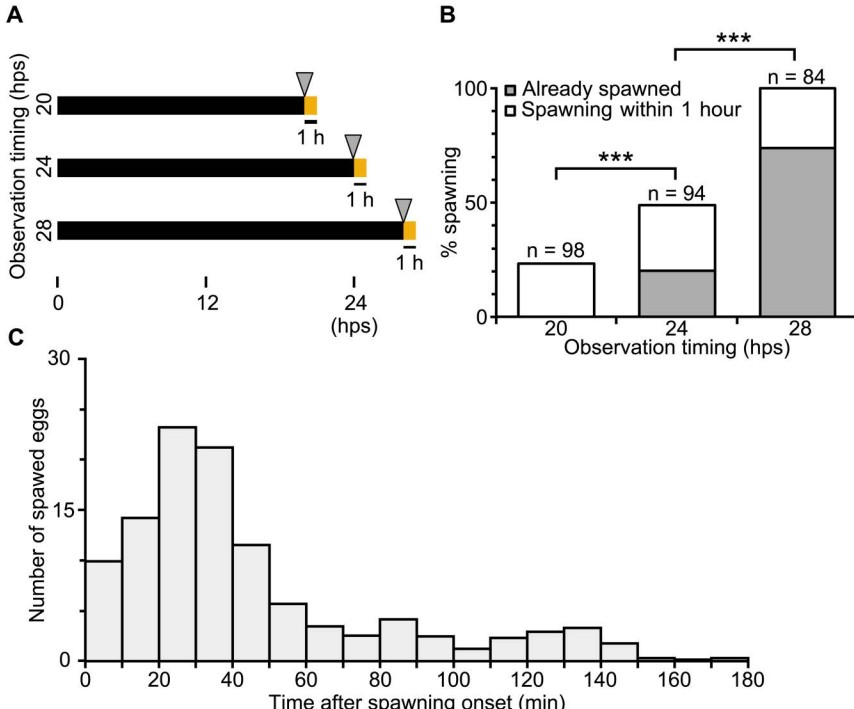

**Fig 4. Asynchronous and spontaneous egg spawning from jellyfish under constant dark. (A)** Experimental design to count spawning from jellyfish maintained in complete dark after the last spawning (LS). Jellyfish maintained in the dark were exposed to light at 20, 24, or 28 hps, and spawning before or one hour after light exposure was scored. **(B)** Percentage of jellyfish that had already started to spawn prior to light exposure (gray box) and those that started one hour after light exposure (white box). *n* = number of jellyfish. **(C)** Individual eggs released from jellyfish (*N* = 10) maintained in constant dark, counted every 10 min after the first egg release from each jellyfish. Spawning observation data are available at https://doi.org/10.1101/2025.05.05.651927.

ovulation event, egg release continued for at least 3 hours (Fig 4C). Overall, these observations show that ovulation is asynchronous and takes longer when *Clytia* sp. IZ-D jellyfish are maintained in a dark environment compared to those under light–dark cycles or constant light. This suggests that light plays a second role in maintaining a synchronous circadian spawning cycle, in addition to modulating ovulation timing through light stimulus.

## A threshold MIH concentration dictates the timing of maturation

We speculate that the quasi-circadian spawning of *Clytia* sp. IZ-D jellyfish might be achieved independently of a canonical circadian clock mechanism, through minor modifications of the opsin-induced MIH release and response mechanisms described previously in *C. hemisphaerica* [13,29]. Under this hypothesis, spawning time could depend on two limiting factors: the accumulation of released MIH in the gonad and the competence of individual oocytes (or inactive sperm) to respond to it. Spawning would occur once both factors exceed critical thresholds, which in female *Clytia* sp. IZ-D jellyfish would be at around 22 hps, resulting in spawning at around 24 hps. To explore this hypothesis, we first determined the effective dose of MIH required to trigger maturation. Fully grown oocytes (stage III; S2 Fig) were isolated from jellyfish cultured in the standard L12/D12 cycle at 22 hps (12 hours after the light stimulus, just before the anticipated timing of MIH-triggered oocyte maturation). Then we examined their response to synthetic WPRPamide, one of the MIH peptides functional in both *C. hemisphaerica* and *Cladonema pacificum* (Fig 5A). The timing of events was compared with that of oocytes within the intact ovary, where Germinal Vesicle Breakdown (GVBD, indicating entry into first meiotic M phase) occurred between 22 and 22.5 hps, and second polar body formation (IIPBF), indicating completion of meiosis, within 90 min (Fig 5B). WPRPamide at concentrations of $10^{-8}$ M or higher consistently triggered oocyte maturation in isolated oocytes, whereas at $10^{-9}$ M only a very low proportion of treated oocytes matured (Fig 5C). These results imply that oocytes receive an MIH signal around 22 hps, initiating the maturation and spawning process.

Two further observations support the conclusion that the effective MIH concentration acting in vivo is equivalent to $10^{-8}$ M WPRPamide applied in external seawater. Firstly, the sensitivity of *Clytia* sp. IZ-D oocytes to WPRPamide was similar to that of *C. hemisphaerica* and *Cladonema pacificum*, where WPRPamide induced oocyte maturation at $10^{-7}$ M but not at $10^{-9}$ M or less [17]. Secondly, the time required to complete oocyte maturation decreased when increasing WPRPamide concentrations (Fig 5D). Notably, the duration from GVBD to first polar body formation (IPBF) and from IPBF to IIPBF shortened with higher peptide concentrations (S3 Movie). Thus, the average duration from GVBD to IIPBF following treatment with $10^{-8}$ M WPRPamide (80 min) was closer to that observed during light-induced oocyte maturation in the gonad ($90 \pm 10$ min) than to that induced by $10^{-5}$ to $10^{-7}$ M WPRPamide (60–70 min).

## Oocyte competence to respond to MIH progressively develop from 18 hps

We next examined the progression of oocytes through the final stages of growth and their competence to respond to WPRPamide when isolated at successive times from 17 to 22 hps. *Clytia* female gonads contain oocytes at different stages, with populations of vitellogenic oocytes progressing through growth stages I to III each day [26,30]. We focused on late stage II and stage III ("fully grown") oocytes, both characterized by the peripheral positioning of the oocyte nucleus (germinal vesicle [GV]). They can be distinguished by the presence or absence of nucleoli (S2 Fig). In *Clytia* sp. IZ-D female jellyfish maintained at 21°C, late stage II oocytes were first detected at 17 hps, and stage III oocytes at 18 hps (S2 Fig). At 17 hps, all oocytes exhibited nucleoli, indicating late stage II, while by 18 hps, most of the oocytes had lost their nucleoli, reaching stage III (Fig 5E and 5F). To compare the competence of stage II versus stage III oocytes to respond to MIH, we isolated oocytes at successive times and treated them with WPRPamide (Fig 5G and 5H). Oocytes isolated at 17 hps (i.e., all at late stage II) failed to respond to WPRPamide at any concentration. Oocytes isolated at 18 hps (S2 Fig) underwent GVBD following treatment with high concentrations ($10^{-5}$ M) of WPRPamide but did not respond to $10^{-8}$ M (Fig 5H). Oocytes gradually became responsive from 20 hps onwards, such that virtually all oocytes responded to $10^{-8}$ M MIH at 21–22 hps, shortly before the onset of maturation under standard L12/D12 conditions. The speed of maturation

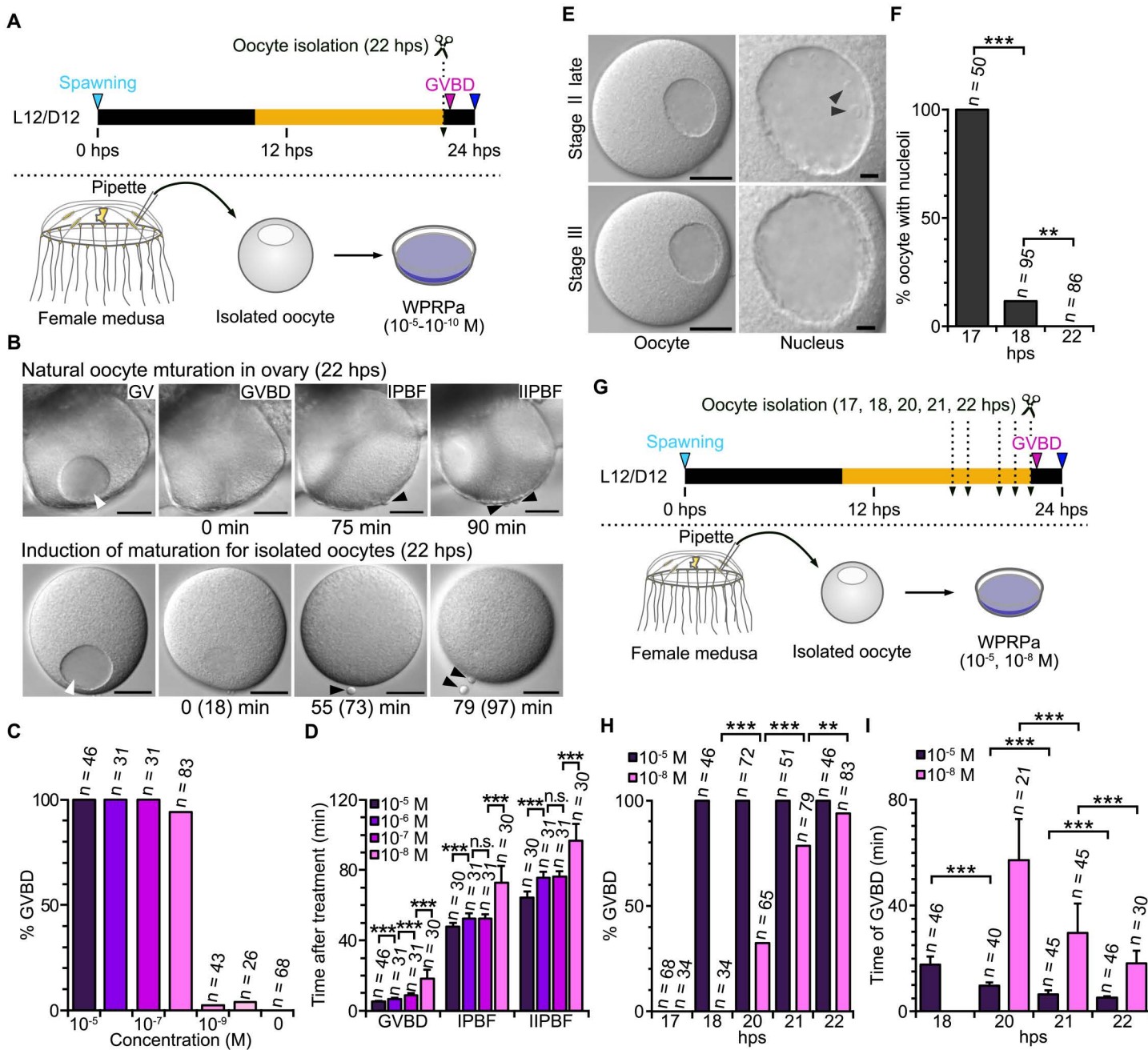

**Fig 5. Oocyte development and acquisition of MIH-reactive competence (MRC).** (A) Experimental procedure. Oocyte isolation at 22 hps, just before endogenous onset of oocyte maturation (meiotic resumption), followed by in vitro induction of maturation with WPRPamide, employed in (B–D). (B) Stages of oocyte maturation in the gonad of jellyfish under L12/D12 cycle and in vitro treated with WPRPamide ($10^{-8}$ M). Times from GVBD are indicated below each panel. Numbers in parentheses show the time counted from WPRPamide addition. (C) Percentage of isolated oocytes that underwent GVBD upon WPRPamide treatment. (D) Time from WPRPamide addition to GVBD, IPBF, and IIPBF. (E) Characteristic morphologies of late stage II oocytes (17 hps) and stage III (18 hps), with a higher magnification of the GV on the right. Stage III is defined by the absence of nucleoli, labeled by triangles. See also S2 Fig for oocyte stages and their development over time. (F) The proportion of grown oocytes with nucleoli, which are indicative of stage II. Absence of nucleoli is characteristic to stage III. (G) The experimental procedure of oocyte isolation at different stages (17–22 hps) and induction of maturation by WPRPamide treatments used in (H) and (I). (H) Percentage of oocytes that underwent oocyte maturation in response to high ($10^{-5}$) and low ($10^{-8}$ M; likely equivalent to the endogenous MIH signal) WPRPamide, after isolation at different stages. (I) Time required from adding WPRPamide to seawater to GVBD. Time required for GVBD following $10^{-8}$ M WPRPamide treatment progressively decreased between 20 and 22 hps. Data for the responses of isolated oocytes to WPRPamide, nucleoli disappearance, and timing of GVBD and polar body formation are available at https://doi.org/10.1101/2025.05.05.651927.

(measured as time from MIH treatment to GVBD) progressively decreased with oocyte age for a given WPRPamide concentration (Fig 5I). Taken together, these results from MIH treatments of isolated oocytes allow us to define the capacity of stage III oocytes to respond to $10^{-8}$ M WPRPamide, which mimics the effective endogenous MIH concentration (see above), as MIH-reactive competence (MRC). The timing of MRC acquisition (i.e., 22 hps) coincides with the endogenous onset of oocyte maturation (22–22.5 hps).

## Oocyte development and MRC acquisition are affected differently by light

To distinguish the respective roles of MRC acquisition and oocyte development in the timing of oocyte maturation and spawning, we tested the MRC of oocytes isolated from jellyfish with culture conditions changed to earlier light stimuli (L16/D8, Fig 6A–6D), constant light (Fig 6E–6H) or constant dark (Fig 6I–6L) after previous spawning under L12/D12. Under these conditions, natural ovulation occurs at 22 hps, 20 hps and asynchronously varying from 20 to 28 hps, respectively. A majority of stage II-ocytes in jellyfish shifted to L16/D8 and constant light conditions lost nucleoli, thus reaching stage III, between 17 and 18 hps (Figs 6B, 6F, and S2). The timing of the transition from late stage II to stage III was thus equivalent to that observed under L12/D12 conditions (Fig 5E and 5F). Most oocytes isolated as early as 20 and 18 hps from L16/D8 and constant light jellyfish responded to $10^{-8}$ M WPRPamide (Fig 6C and 6G, 85% and 92% respectively), indicating MRC, in contrast to the L12/D12 cycle (Fig 5H) in which MRC was acquired at 21–22 hps. These timings of MRC acquisition assayed in isolated oocytes correlate with the onset of oocyte maturation in intact jellyfish (Table 1). They confirm that MRC acquisition can be uncoupled from morphological transition to stage III and, importantly, rules out the possibility that it is merely a function of growth time since the last spawning. Instead, these results suggest that MRC acquisition is influenced by gonadal factors under the control of the light–dark cycle. Under constant dark conditions, unlike other conditions bearing a certain light period, morphologically defined oocyte development was disrupted. Many of the grown oocytes isolated at 18 hps (32%) had lost their nucleoli, while as many as 44% were still at late stage II even at 24 hps (Fig 6J). Also the MRC acquisition timing was variable. Fourteen percent of oocytes at 18 hps exhibited MRC, while this increased only to 53% at 24 hps, which coincides with the asynchronous oocyte release from jellyfish under constant dark condition (Fig 4). Overall, the timing of MRC acquisition exhibited a good correlation with the endogenous onset of oocyte maturation (GVBD) timing. On the other hand, MRC and arrival at stage III were not coupled, with each influenced by light in distinct ways.

## Acquisition of MIH-reactive competence by stage II oocytes at 17 hps relies on prior light exposure

To test whether oocytes require the gonadal environment to acquire MRC, we isolated late stage II oocytes at 17 hps from jellyfish maintained under a standard L12/D12 cycle. These oocytes were then cultured in seawater (Fig 7A). Oocytes isolated earlier could not reach stage III when cultured in vitro, presumably because gonad somatic cells are necessary to support aspects of earlier development. Isolated late stage II oocytes lost nucleoli, indicating progression from late stage II to stage III, within 45 min after isolation (Fig 7B and S4 Movie). As described above, late-stage II oocytes immediately after dissection could not initially respond to MIH (Fig 5H). After one hour of incubation in seawater, they became reactive to $10^{-5}$ M WPRPamide (Fig 7C) but not to $10^{-8}$ M. After four to five hours of incubation, they became reactive to $10^{-8}$ M WPRPamide, with GVBD occurring with the endogenous timing (22 hps, Fig 7C and 7D), and thus can be considered to have acquired MRC. This timing corresponds to 21 and 22 hps. We conclude that the timing of MRC acquisition is not significantly different between oocytes isolated at 21–22 hps and those isolated 17 hps then cultured in vitro. Similarly, in vitro cultures of oocytes isolated at 17 hps from jellyfish shifted to shorter dark periods (L16/D8) and constant light acquired MRC 3 and 1 hours after isolation, respectively (Fig 7E–7J), again concurrently with the normal GVBD timing of oocytes in the gonad (20 and 18 hps, respectively). Oocytes isolated at 17 hps from jellyfish shifted to the constant dark condition gradually acquired MIH but did not exceed 50% even after 7 hours of incubation (Fig 7K–7M). This indicates that

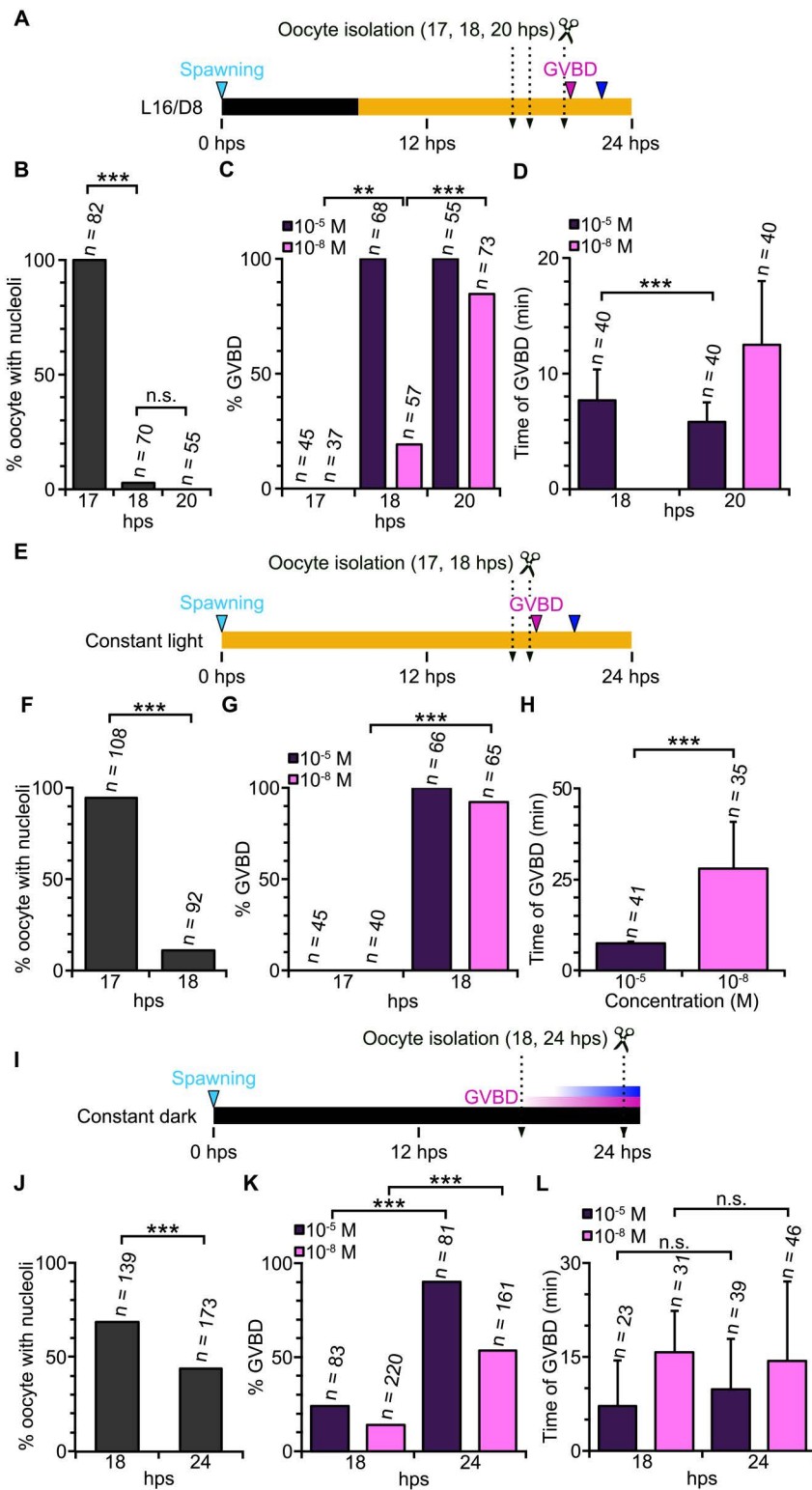

**Fig 6. Timing of MRC acquisition of isolated oocytes mirrors that of spawning. (A–D)** Reactivity of isolated oocytes to high ($10^{-5}$ M) and low ($10^{-8}$ M) GLWamide and time for GVBD when light–dark cycle was changed from L12/D12 (Fig 5G–5I) to L16/D8 **(A–D)**, constant light and **(E–H)** and constant dark **(I–L)**. **(A, E, I)** timing of oocyte isolation indicated by scissor icons, between 17 hps (arrival to stage II) and spontaneous GBVD timing in the

gonad (magenta triangle). The onset of the spawning is indicated with blue triangles. **(B, E, J)** Proportion of isolated grown oocytes containing nucleoli, indicative of stage II. **(C, G, K)** Percentage of oocytes that underwent GVBD following WPRPamide treatment. **(D, H, L)** Time from WPRPamide addition to GVBD for oocytes isolated at 18 hps. Data for the responses of isolated oocytes to WPRPamide, nucleoli disappearance, and timing of GVBD and polar body formation are available at https://doi.org/10.1101/2025.05.05.651927.

**Table 1. Summary of WPRPamide responses of oocytes isolated at different times or following in vitro culture after isolation at 17 hps.**

| Condition | Normal GVBD timing (hps) | Response of isolated oocytes to 10$^{-8}$ M WPRPamide: isolated at each time indicated hps* | | | | | | Response of oocytes to 10$^{-8}$ M WPRPamide, isolated at 17 hps and in vitro cultured for the number of hours shown (hps equivalent in parentheses)* | | | | | |
|---|---|---|---|---|---|---|---|---|---|---|---|---|---|
| | | 17 | 18 | 20 | 21 | 22 | 24 | 0 (17) | 1 (18) | 3 (20) | 4 (21) | 5 (22) | 7 (24) |
| L12/D12 | 22+ | − | − | ± | ++ | ++ | | − | − | | ± | ++ | |
| L16/D8 | 20+ | − | ± | ++ | | | | − | ± | ++ | | | |
| Constant Light | 18+ | − | ++ | | | | | − | ++ | | | | |
| Constant Dark | Variable (20–27) | | ± | | | | ± | − | ± | | | | ± |

*++ indicates 60% or more oocytes underwent maturation; ± indicates maturation observed in less than 60% of oocytes; − indicates no maturation.

oocytes are autonomously programmed to acquire MRC autonomously by 17 hps, depending on the preceding light–dark cycle.

Together, our observations suggest that attainment of stage III and MRC acquisition are controlled by light in two distinct mechanisms. Early light exposure prior to 17 hps promotes synchronous development of oocytes to reach stage III, characterized by nucleoli loss. In contrast, the timing of MRC acquisition after reaching stage III is influenced by the previous dark-light cue, potentially delaying spawning by up to four hours (Fig 8).

## Discussion

From the Pacific coast of Japan, we identified a new species of hydrozoan jellyfish *Clytia* sp. IZ-D, which releases gametes at dusk in the natural habitat. Through observation of egg release, we revealed that *Clytia* sp. IZ-D possesses an autonomous circadian rhythm of synchronous gamete release every 20 hours at 21°C under constant light conditions (Fig 8A). The periodic cycle length is altered by the temperature. Under light–dark cycle, synchronous ovulation becomes entrained to a 24-hour period, in which gamete release occurs synchronously 14 hours after the preceding dark-to-light transition at 21°C (Fig 8B), which requires a minimum of four hours of dark followed by light illumination. The regulation of gamete release by this dark-to-light cue is not entirely determinative but is constrained within a range of 20–24 hours from the previous spawning event. In females, the earlier limit is defined by a minimal period of 18 hours required for development for the daily cohort of growing oocytes. Thus, in this species, the light stimulus entrains the autonomous circadian ovulation cycle. This is the first observation that hydrozoan jellyfish display an autonomous circadian rhythm controlling precise and synchronized gamete release. Under constant darkness, autonomous egg maturation and release occur asynchronously, starting from 20 hours after the previous ovulation (Fig 8C). Light therefore plays an additional, permissive role in maintaining the autonomous quasi-circadian rhythm.

Our observations show that key determinants of the autonomous ovulation cycle are oocyte growth and the acquisition of maturation competence. The shortest ovulation interval (20 hours at 21°C) is specified by the time required for growing oocytes to develop to stage III, the stage at which they are ready to undergo meiotic maturation, occurring at 18 hps. The actual timing of ovulation correlates with the timing of the oocytes to acquire the competence to respond to 10$^{-8}$ M WPR-Pamide in the external seawater (MRC). This concentration simulates the effective endogenous MIH in the gonad, based

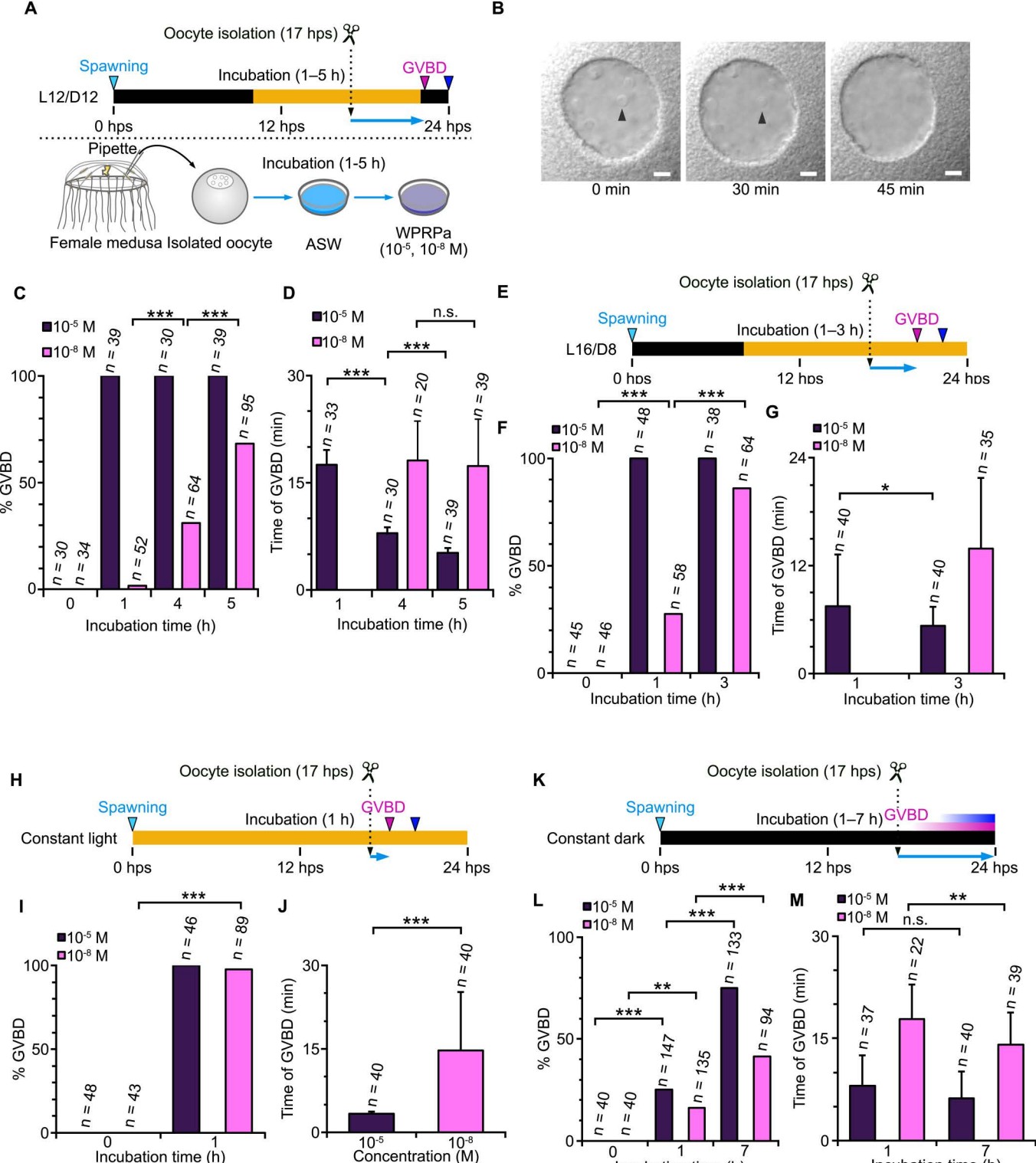

**Fig 7. Late stage II oocytes autonomously develop to stage III and acquire MRC in vitro. (A)** Experimental procedure. Late stage II oocytes isolated at 17 hps from the gonad of jellyfish under L12/D12 cycle were incubated in seawater for up to 5 hours (corresponding to 22 hps, shortly before the endogenous GVBD timing in gonad), then treated with WPRPamide. **(B)** GV morphology during culture. Arrowheads indicate nucleoli, which disappear

within 45 min. **(C)** Percentages of oocytes that underwent oocyte maturation with WPRPamide, before (0 hour) and after (1–5 hours) in vitro culture. **(D)** Time from WPRPamide addition to GVBD. **(E–G)** In vitro culture experiment for jellyfish shifted to L16/D8 (shorter dark period), which spawn at 22 hps, **(H–J)** constant light; spawn at 20 hps and **(K–M)** constant dark; spawn asynchronously between 20 to 28 hps. Data for the responses of isolated oocytes to WPRPamide and polar body formation are available at https://doi.org/10.1101/2025.05.05.651927.

on the estimation derived from the duration between GVBD and PBFI/PBFII. Assuming this level is present in the gonad from at least 18 hps (see below), oocyte growth and acquisition of MRC would be sufficient to determine the next oocyte maturation and ovulation schedule and thus to account for the autonomous circadian gamete release observed in constant light in *Clytia* sp. IZ-D.

## How conserved MIH-induced oocyte maturation process exhibits circadian cycle

MIH-induced oocyte maturation has been demonstrated in a number of hydrozoan species, including *C. hemisphaerica*, which is closely related to *Clytia* sp. IZ-D, and *Cladonema pacificum*. These species utilize nearly identical peptide sequences, respectively, RPR(P/A)amide and (W/R)PRPamide, as the endogenous MIH [17]. In *C. hemisphaerica*, neurosecretory cells in the gonad ectoderm expressing an opsin, CheOpsin9, release MIH upon light stimuli. Oocytes receive MIH and undergo maturation (completion of meiosis), beginning with GVBD about 20 min after light stimulation or MIH addition to isolated oocytes. cAMP/PKA signaling acts downstream of G-protein-coupled MIH-R [12,29]. This process completes within 90–120 min [26]. The timeline from GVBD to the completion of meiosis is similar in *Clytia* sp. IZ-D and suggests that MIH signaling in this species is also received and transduced by oocytes approximately two hours before ovulation. We do not yet know when and how MIH is released into the gonadal extracellular space in *Clytia* sp. IZ-D. It seems unlikely that MIH is released as a discrete pulse exactly 12 hours after light stimulation. Nor is it likely that this process is governed by the canonical metazoan circadian oscillation mechanisms based on BMAL1(CYCLE)/CLOCK and PER/CRY/TIM as BMAL1 and CLOCK appear to have been lost in a common hydrozoan ancestor [25]. Furthermore, the autonomous spawning cycle length drifted extensively depending on temperature, contrasting with authentic circadian clock systems that commonly exhibit robust temperature-compensation properties [27,28,31,32]. We thus propose that the characteristic spawning cycle of *Clytia* sp. IZ-D, rather than relying on conventional circadian oscillator mechanisms, involves gonad neurosecretory cells expressing opsins and mediating MIH release, as in *C. hemisphaerica*, but operating with distinct kinetics. One possibility is that MIH is slowly released into the extracellular space during light conditions from MIH-expressing neurosecretory cells, and that oocytes only respond once they have activated sufficient G-protein-coupled MIH-R [29] on the oocyte surface to elicit the downstream response. Our current hypothesis is that the crucial elements determining spawning timing are oocyte growth and acquisition of MRC (Fig 8), with MIH release occurring slowly and/or constitutively under the presence of light.

## Oocyte growth and maturation competence as key timer components

As outlined above, our observations suggest that the competence of oocytes to respond to MIH (MRC) serves as a key control point in defining the timing of oocyte maturation. Our findings highlighted two light-dependent processes: the transition of oocytes to stage III and the acquisition of MRC. The attainment of stage III, characterized by the loss of nucleoli and readiness for meiotic maturation and likely corresponding to the partial chromosome condensation at diakinesis stage [30], occurs at 18 hps in all conditions except for continuous dark (Fig 8). This partly explains why ovulation does not occur at 20 hps or earlier and indicates that light exposure during the growth period is necessary for synchronous oocyte development to stage III. Similar minimum intervals of spawning constrained by the oocyte growth have also been observed in other hydrozoan species. Future studies in other species will test whether exposure to light favoring synchronous oocyte development is a common feature in hydrozoans. The second control point, the acquisition of MRC, is influenced by the timing of a preceding dark–light transition. The response of oocytes isolated from gonads to MIH demonstrates that MRC acquisition timing, which varies from 18 to 22 hps, is autonomously programmed in the oocytes by 17 hps, depending on the timing of the dark–light

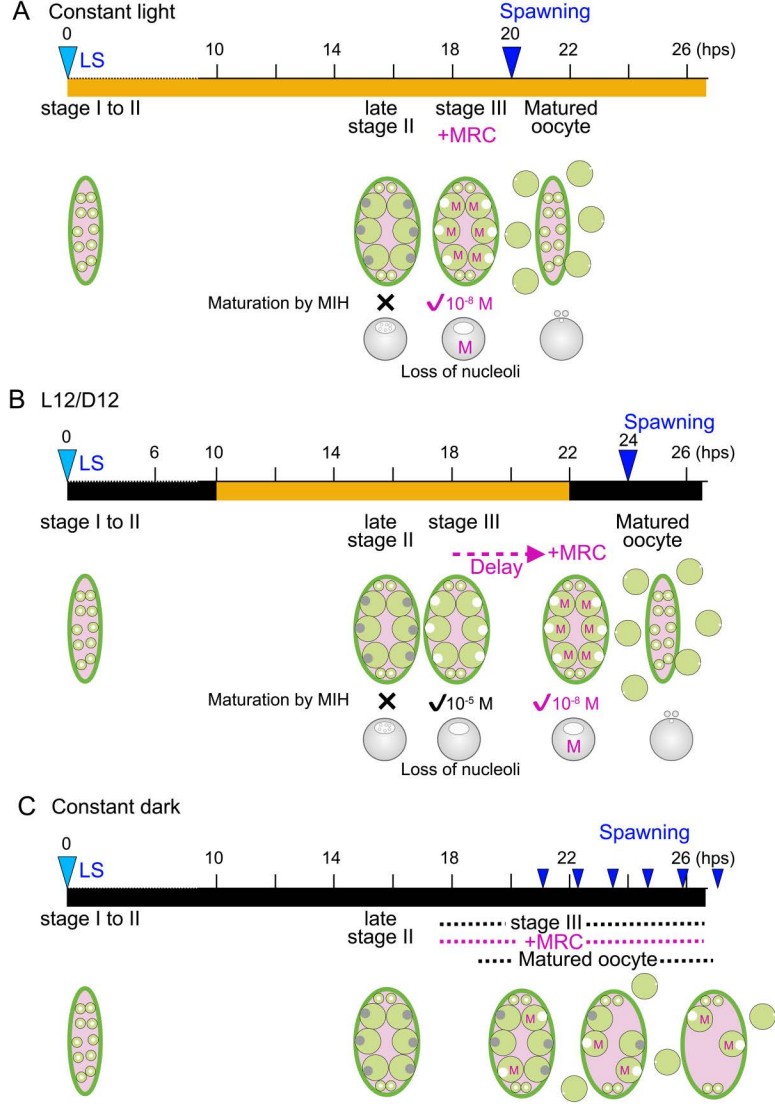

**Fig 8. Schematic comparison of the timing of oocyte development, MRC and spawning under three light/dark regimes. (A)** In constant light, synchronous ovulation occurs every 20 hours. Both stage III arrival and MRC occur at 18 hps, followed by a period of two hours for the completion of meiotic maturation before spawning. **(B)** Under a L12/D12 cycle, synchronous spawning occurs every 24 hours, 14 hours after the light stimulus. Stage III arrival and MRC occur at 18 and 21–22 hps, respectively, with synchronous spawning at 24 hps. **(C)**. In constant darkness, spawning is asynchronous. MRC, defined as the ability to respond to $10^{-8}$ M exogenous WPRPamide, deduced to match the endogenous MIH level required to trigger spawning. Previous spawning (LS) is indicated with light blue triangles. Synchronous (A and B) and asynchronous (C) spawning events are shown with dark blue triangles.

stimulus before this stage. One possible model that can explain the 14-hour delay from the onset of the light period to oocyte maturation is that oocytes acquire MRC progressively over time, and that a transition to light following four or more hours of darkness stalls this process for several hours (Fig 8B). This MRC acquisition process appears to be controlled independently from the transition from late stage II to III, which requires an exposure to light (Fig 8). This may explain why ovulation timing varies among oocytes in the same gonad, up to 40–60 min under L12/D12 cycle conditions and constant light (Figs 2H and 3B). Once MIH activity in the gonad reaches the effective concentration, equivalent to $10^{-8}$ M of WPRPamide applied externally, individual oocytes will react to it as soon as they acquire MRC.

It will be fascinating to address the molecular basis of the light modulation of gamete development and MRC in future studies. One possible model is that the gradual increase of MRC in oocytes may be stimulated by continuous weak signaling from light-sensitive neurosecretory cells in the gonad ectoderm, contrasting with the immediate release of MIH triggered by CheOpsin9 activation to stimulate immediate oocyte maturation upon light stimulus in *C. hemisphaerica*. The molecular nature of such a continuous signal could potentially be mediated by MIH itself and/or other neuropeptides. One possibility is that the cAMP signaling downstream of MIH/MIH-R, which induces oocyte maturation, also promotes oocyte development. Under this hypothesis, light-induced delay of MRC acquisition (e.g., delay of MRC/spawning under L12/D12 cycle compared to continuous light) could be explained either by attenuation of the slow MIH release several hours after light stimuli, or by the release of antagonistic neuropeptides upon light stimulus. In line with this, spawning is delayed only when light stimulus is applied between 6 and 10 hps, raising an interesting question of which oocyte development process during the prolonged prophase I arrest are modulated by the light stimulus.

Our speculative models to explain the unusual quasi-circadian spawning cycle, in *Clytia* sp. IZ-D remains largely open for testing. Slow MIH release alone is unlikely to explain ovulation timing, as it would require implausibly precise control to reach a receptor-binding threshold with such accuracy in the absence of a timer. An entirely plausible possibility is that a circadian oscillator is involved. While the BMAL1/CLOCK system is absent in hydrozoans, it has been shown that this ancient circadian mechanism can be rewired during evolution by co-opting alternative components [33]. Orthologues of some key components of the BMAL1/CLOCK transcriptional-translational feedback loop, notably Cryptochrome (CRY1) [34,35] and its downstream PAR/bZIP regulator HLF and TEF [36] are present in the *C. hemisphaerica* genome. During mammalian melanogenesis, CRY1 inhibits the cAMP/PKA pathway [37]. Another potential mechanism to explore is thus that *Clytia* Cryptochrome similarly down-regulates the MIH-R/PKA pathway in developing oocytes following light stimulation to explain the observed delay in MRC acquisition.

Comparable mechanisms likely account for the timing of male gamete release. In *Clytia* IZ-D male jellyfish, sperm release occurs slightly earlier (approximately 13 hours after the light stimulus) than female ovulation (S1C Fig). Little is known about how sperm release is controlled. In *C. hemisphaerica*, spermatozoa accumulated in male gonads have completed meiosis and are activated by MIH to become motile and be released from the gonad [30]. In the coral *Astrangia poculata,* sperm activation is induced by cAMP signaling [38], which is likely the second messenger of the seven transmembrane MIH receptor MIH-R, expressed during spermatogenesis as well as oogenesis in *Clytia* [29].

## Evolutionary and ecological advantages

For marine species that externally release and broadcast gametes, timing control is crucial to successfully achieving sexual reproduction. In both *C. hemisphaerica* and *Clytia* sp. IZ-D, gamete release is coordinated between sexes, with male sperm release starting slightly earlier than female ovulation [30]. An earlier onset of sperm release, allowing wider sperm dispersal, may be an adaptive strategy for successful fertilization under sperm competition [39] in sparsely distributed jellyfish populations. In *C. hemisphaerica*, fertilization success requires gamete mixing within one hour after egg release. Sunrise and sunset are the few environmental cues that allow animals at a distance to synchronize gamete release. Most hydrozoan jellyfish documented release gametes within a few hours of light stimulation [10–12,16]. Multiple corals in the same regions will release gametes simultaneously, risking hybridization [40]. Changing spawning times may confer an advantage by avoiding cross-species fertilization, and a genetic change that alters the timing of gamete release within a population could reproductively isolate it from the others. This, in turn, could lead to the eventual establishment of a new species [41]. We identified at least two *Clytia* species, *Clytia* sp. IZ-D and *Clytia* sp. IZ-C in the bay of Izuashima Island (Fig 1A and 1D). The adaptation of gamete release timing to dusk, potentially through the innovation of the alternative circadian clock, might have allowed the speciation of *Clytia* sp. IZ-D from the common ancestor shared with other *Clytia* species. Another potential ecological advantage of an autonomous circadian cycle is the robustness against seasonal climate instability. *Clytia* sp. IZ-D jellyfish can continue synchronized gamete release even if the sunlight does not follow

a regular 24-hour cycle, for example, during the seasonal typhoons in summer and autumn in this region, or under the influence of artificial light sources.

## Materials and methods

### Jellyfish collection

Medusae of the genus *Clytia* (*Clytia* sp. IZ-D, *Clytia* sp. IZ-C) were collected through a series of regular sampling at Terama Fishery Port in Izushima Island (Miyagi Prefecture, Japan) from July to October 2023. Collection was performed using the jellyfish net described previously [42] , consisting of a 1 mm nylon mesh filter attached to a 300 mm diameter metal ring frame. The net was cast from the shore and slowly dragged at a constant speed to target plankton located less than 2 m below the sea surface, with sampling repeated over several hours. Collected jellyfish were transferred into a stainless-steel cooking tray with seawater and identified morphologically.

### Jellyfish culture and polyp colony establishment

We used laboratory-reared jellyfish for all experiments. Artificial seawater (ASW: SEALIFE salt, 35 g/L ($\approx$ 30 ppt, Marinetech, Japan) was used for culture. The male and female *Clytia* sp. IZ-D jellyfish identified in October 2023 were placed together in a dish to obtain fertilized eggs. The resulting planula larvae were transferred to a new polypropylene dish (60 mm diameter, 35 mm height), where they underwent natural metamorphosis and formed polyp colonies. Colonies were subsequently transplanted to individual dishes and maintained in a larger tank with water flow maintained by fine-bubble aeration. Newly budded medusae were collected from the polyp colony dishes and reared separately in other dishes filled with ASW. Dishes were covered by sealing lids and stored in a 21°C incubator. Under these culture conditions, medusae began releasing eggs and sperm within 2 weeks. Jellyfish and polyp colonies were fed live *Artemia salina* nauplii once every day or two, and ASW was changed after feeding.

### Local climate data

Average monthly water temperature data from Enoshima Island station (8 km southwest of Izushima Island) were provided by the Miyagi Prefecture Fisheries Technology Institute (https://tohokubuoynet.myg.affrc.go.jp/Vdata/). Sunlight data for Sendai city (65 km west of Izushima) for 2024 was obtained from the Ephemeris Computation Office of the National Astronomical Observatory of Japan (https://eco.mtk.nao.ac.jp/koyomi/dni/dni04.html.en).

### Phylogenetic analysis of 16S rRNA sequences

DNA was extracted from medusae using the NucleoSpin Tissue XS (MACHEREY-NAGEL) according to the manufacturer's protocol and used as a PCR template. PCR was conducted using KOD One PCR Master Mix (TOYOBO), with PCR primers 16S-F1 (5′-ACGGAATGAACTCAAATCATGTAAG-3′) and 16S-R1 (5′-CCTTTTGTATAATGGATTTACAAG-3′). PCR cycle was as follows: initial denaturation at 98°C for 1 min, followed by 35 cycles of 10 s at 98°C, 5 s at 50°C or 52°C, and 5 s at 68°C, with a final extension at 68°C for 1 min. The PCR products were purified using Nucleospin Gel and PCR Clean-up kit (MACHEREY-NAGEL) and sequenced at Eurofins Genomics (Tokyo, Japan). The sequences obtained were aligned using MUSCLE in the MEGA 11 software for macOS [43,44]. The alignment data of 16S rRNA, including their accession numbers, is provided in S1 File in FASTA format. Phylogenetic trees were generated using the maximum-likelihood (ML) method with 1,000 bootstrap replicates, based on the K2P model. The sequence of *Cladonema pacificum* (AB720901.1) for 16S rRNA was used as the outgroup. Bootstrap values greater than 50% are shown above the branches as node support values in Fig 1D.

### Adjustment of light–dark cycle and observation of spawning timing

*Clytia sp* IZ-D D-A1 and D-A3 strain jellyfish, less than three months after liberation from the polyp colonies, were used in this study. Mature jellyfish in culture dishes were placed in a 21°C incubator under a light–dark cycle controlled by a

programmable timer (Koizumi Computer or Ohm Electric) and illuminated with white LED light (Yazawa LE3WH, 0.2 W). Under standard culture/growth conditions, jellyfish were maintained on a 12-hour light and 12-hour dark cycle (L12/D12) at 21°C. After the last spawning event (marked in LS in the figures), each jellyfish was transferred to an individual well of a 6-well plastic dish (AsOne VTCP-6) and maintained in another incubator (Mitsubishi Electric CN-40A) shifted to the experimental light–dark cycle conditions. The timing of egg and sperm release for each jellyfish was defined by the initiation of release and measured through continuous observation using a stereomicroscope (Leica M165C) or an inverted microscope (Nikon Eclipse TE300). In the case of spawning during the dark period, where observation under a light microscope might disturb spawning, we first estimated the approximate timing of spawning by pilot experiments conducted without light. Precise observations were then made under light microscopes starting 40 min before the anticipated spawning time, at which point oocyte maturation had already commenced, thus minimizing any impact on oocyte maturation timing. We also confirmed that the interruption of dark periods by this observation did not affect the ovulation timing on the following day. For jellyfish maintained under constant dark conditions, where spawning timing was highly variable among jellyfish and oocytes, and where the light exposure would interrupt the dark environment, jellyfish were sampled at different time points, and the spawning by the time of exposure and in the following hour was observed. See also the main text.

## Oocyte isolation from the gonad and in vitro maturation

Jellyfish were anaesthetized in ASW containing $Mg^{2+}$ (a 1:1 mix of 0.53 M $MgCl_2$ and ASW) during manipulation. Oocytes were isolated from the gonad of female jellyfish under an upright microscope (Nikon Eclipse 80i) by aspirating with a microcapillary pipette, prepared by pulling a Pasteur pipette under a Bunsen burner flame and cutting the tip to an opening of 0.2 mm. Immediately after isolation, oocytes were staged based on nuclear morphology and the presence and number of nucleoli (see S2 Fig for staging criteria). Small oocytes (earlier than early stage II) were excluded. Isolated oocytes were maintained in plastic Petri dishes containing ASW. The peptide $WPRP-NH_2$ (WPRPamide), the most potent isoform of the endogenous MIHs identified in *C. hemisphaerica* [17] was synthesized by Genscript, dissolved into $H_2O$ at $10^{-3}$ M and stored at −20°C. Before use, it was diluted and added to the culture ASW at concentrations ranging from $10^{-5}$ to $10^{-8}$ M. Isolated oocytes were transferred to wells of a 24-well plate (Falcon 353047) containing the ASW supplemented with WPRPamide. The timing of oocyte maturation events, notably GVBD, IPBF, and IIPBF, was recorded through observation under the inverted microscope. Completion of oocyte maturation was defined by IIPBF. Experiments with isolated oocytes were performed in a room maintained at approximately 21°C.

## Ethics statement

Wild jellyfish used for establishing the laboratory strains were collected from coastal waters in Izushima, Miyagi Prefecture, Japan, with permission from Terama Fishery Port local office of the Miyagi Prefectural Federation of Fisheries Cooperative Associations. The sampling did not involve any protected or endangered species and was conducted outside the marine protected zones of the Sanriku Fukko National Park. No specific ethical approval was required for collecting invertebrates not listed as nationally endangered species (https://www.env.go.jp/nature/kisho/domestic/list.html) under Japanese law; however, all procedures complied with institutional and national guidelines for the ethical use of wildlife in research.

## Supporting information

**S1 Fig. Sperm release from male jellyfish is coordinated with female ovulation. (A)** Isolated male gonads that release sperm. Scale bar: 0.5 mm. **(B)** A high magnification image of a sperm. Scale bar: 10 μm. **(C)** Comparison of male and female spawning time measured from the onset of light stimuli. Sperm release occurred earlier (13.0 ± 0.3 hours) than egg spawning (13.9 ± 0.3 hours). Sperm release timing is defined by its onset from a jellyfish, as it continues up to 1 hour. Bars indicate standard deviation; $n$ = number of jellyfish. (Welch's test, *** $p \le 0.001$) (D) Sperm release also exhibits

regular spawning at dusk under a light–dark cycle and autonomously under constant light. Egg spawning and sperm release observation data are available at https://doi.org/10.1101/2025.05.05.651927.
(PDF)

**S2 Fig. Oocyte development. (A)** DIC images of oocytes and their nuclei at different stages. The staging criteria were based on Amiel and colleagues [26] for *Clytia hemisphaerica*, with additional clarification that late stage II is defined by peripheral migration of GV and presence of nucleoli, while stage III is determined by the disappearance of nucleoli. Arrowheads indicate nucleoli. Bars are 50 μm for oocyte panels and 10 μm for nucleus panels. **(B)** Characteristics of oocyte stages and their occurrence after the previous spawning.
(PDF)

**S3 Fig. Development and fertilization rates. (A)** DIC images of each embryonic stage and planula larva. Scale bars: 50 μm. Time after fertilization is indicated in parentheses. **(B)** Fertilization efficiency at different temperatures (chi-squared test, *** $p \leq 0.001$, n.s. $p \leq 0.05$). Fertilization efficiency data are available at https://doi.org/10.1101/2025.05.05.651927.
(PDF)

**S1 Movie. Time-lapse movie showing egg spawning from an isolated female gonad.** The movie corresponds to 50 min in real time. Elapsed time from the onset of spawning is shown as on-screen captions. Spawning continued for approximately 40 min.
(MOV)

**S2 Movie. Time-lapse movie showing sperm release from an isolated male gonad.** The movie corresponds to 50 min in real time. Elapsed time from the onset of spawning is shown as on-screen captions. Sperm release continued for approximately 50 min.
(MOV)

**S3 Movie. Time-lapse sequence of oocyte maturation following in vitro treatment with a high concentration ($10^{-5}$ M) of WPRPamide.** The entire movie corresponds to 60 min in real time. Elapsed time from WPRPamide treatment is shown as on-screen captions. GVBD occurred at 5 min, IPBF at 45 min, and IIPBF at 60 min after WPRPamide treatment.
(MOV)

**S4 Movie. Time-lapse sequence showing nucleolar disappearance in the nucleus of an in vitro cultured oocyte isolated at 17 hps.** The movie corresponds to 60 min in real time. Elapsed time from oocyte isolation is shown in the top-left corner. Nucleoli disappeared after 60 min. The black arrow indicates a nucleolus.
(MOV)

**S1 File. 16S ribosomal DNA sequence in FASTA format used for the phylogenetic analysis (Fig 1D).**
(TXT)

## Acknowledgments

We thank to Terama Fishery Port administration for authorizing jellyfish towing. We would like to thank the three anonymous reviewers for their extensive and constructive comments that have improved this paper.

## Author contributions

**Conceptualization:** Ruka Kitsui, Ryusaku Deguchi.

**Data curation:** Ruka Kitsui, Noriyo Takeda.

**Formal analysis:** Ruka Kitsui.

**Funding acquisition:** Ryusaku Deguchi.

**Investigation:** Ruka Kitsui.

**Resources:** Ruka Kitsui.

**Supervision:** Ryusaku Deguchi, Tsuyoshi Momose.

**Visualization:** Ryusaku Deguchi, Tsuyoshi Momose.

**Writing – original draft:** Tsuyoshi Momose.

**Writing – review & editing:** Evelyn Houliston, Tsuyoshi Momose.

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
