## [Editor Report · Decision Letter 0]

26 Jun 2025

Dear Dr Momose,

Thank you for submitting your manuscript entitled "A light-modulated clock mechanism in a hydrozoan jellyfish that synchronises evening gamete release" for consideration as a Research Article by PLOS Biology and I apologize for the delay in sending you an initial decision. I had wished to discuss your paper with an Academic Editor with relevant expertise, however I was not able to find someone who was available to provide advice over the last week.

In the absence of feedback from an Academic Editor, I have discussed your study with my colleagues on the PLOS Biology editorial team, and I am writing to let you know that we would like to send your submission out for external peer review. I should note that, while we are, in principle, interested in your study, we have yet to make a firm call about whether it offers the level of conceptual advance that we would aim to publish - and so we will be looking for strong support from the reviewers in that regard. I also wanted to mention that given that it is, as yet, unclear what is the "unusual circadian timing mechanism" that regulates the autonomous spawning cycles in this species, we think that your paper would be best suited for consideration as a "Discovery Report", which can be a bit more mechanistically preliminary. We think that your paper will have the greatest chance of a successful outcome in that format.

The main thing to note about Discovery Reports, is that these have a limit of 4 figures. For now, we are OK reviewing the study in its current 8 figure form - but if we decide to move forward with the paper, we will likely ask that you condense the figures accordingly.

Before we can send your manuscript to reviewers, we need you to complete your submission by providing the metadata that is required for full assessment. To this end, please login to Editorial Manager where you will find the paper in the 'Submissions Needing Revisions' folder on your homepage. Please click 'Revise Submission' from the Action Links and complete all additional questions in the submission questionnaire.

Once your full submission is complete, your paper will undergo a series of checks in preparation for peer review. After your manuscript has passed the checks it will be sent out for review. To provide the metadata for your submission, please Login to Editorial Manager (https://www.editorialmanager.com/pbiology) within two working days, i.e. by Jun 30 2025 11:59PM.

Kind regards,

Luke

Lucas Smith, Ph.D.

Senior Editor

PLOS Biology

lsmith@plos.org

---

## [Decision Letter · Decision Letter 1]

4 Aug 2025

Dear Tsuyo,

Thank you for your patience while your manuscript "A light-modulated clock mechanism in a hydrozoan jellyfish that synchronises evening gamete release" was peer-reviewed at PLOS Biology. It has now been evaluated by the PLOS Biology editors, an Academic Editor with relevant expertise, and by several independent reviewers.

In light of the reviews, which you will find at the end of this email, we would like to invite you to revise the work to thoroughly address the reviewers' reports.

As you will see below, the reviewers find the study to be interesting and generally well performed. However, the reviewers have also provided a number of suggestions to strengthen the study further, and we think that their comments should be thoroughly addressed before we can accept your study for publication. While we think that the revision will be relatively straightforward, we are providing a 3 month deadline so that you can provide the analyses requested by reviewer 2, as we think that would be important data to add (if feasible).

As a last editorial note - before review we had suggested the paper should be considered as a Discovery Report, however given the reviewer feedback, we have reconsidered that request. Given that the reviewers appreciate the comprehensive nature of the work, we end up thinking it would be better to leave the study as a full 'research article', so that you don't have to condense any of the figures or move any of the results to the supplement. Therefore, feel free to disregard our previous suggestion to reformat this into a 4 figure report. When resubmitting, we ask that you change the article type again, back to "research article".

Given the extent of revision needed, we cannot make a decision about publication until we have seen the revised manuscript and your response to the reviewers' comments. Your revised manuscript is likely to be sent for further evaluation by all or a subset of the reviewers.

**IMPORTANT - SUBMITTING YOUR REVISION**

*Re-submission Checklist*

*Published Peer Review*

*PLOS Data Policy*

*Blot and Gel Data Policy*

Sincerely,

Luke

Lucas Smith, Ph.D.

Senior Editor

PLOS Biology

lsmith@plos.org

REVIEWS:

Reviewer #1: Review: "A light-modulated clock mechanism in a hydrozoan jellyfish that synchronises evening gamete release." (PBIOLOGY-D-25-01890R1).

The manuscript by Kitsui and colleagues describes a novel rhythmic phenomenon relevant for the biology of hydrozoan jellyfish; but more generally, it is an interesting report for understanding clocks.

In many marine animals with external fertilisation, the timing of gametes release is a life-or-death choice. The survival of the species depends on the ability of eggs and sperms to find each other in the vastity of the sea, and without getting confused with others. Light is an obvious synchroniser. Indeed, light-dark changes, either because of the sun or of the moon cycle, are universal reference points that all individuals can recognise and align to. Moreover, by setting release to occur at a specific phase of the cycle, different species can occupy a unique temporal niche in an otherwise overlapping space, just like booking a table at the restaurant. In the hydrozoan jellyfish Clytia hemisphaerica, a light stimulus triggers the secretion of a maturation-inducing hormone (MIH) from the gonads, which drives the (final) meiotic maturation and the release of the oocytes. Perhaps not surprisingly, the MIH-secreting cells express opsins. In this manuscript the authors describe a novel species, Clytia sp. IZ-D, a close relative of C. hemisphaerica. Interestingly, in this new species, oocytes are released not as a direct response to a light stimulus but according to what it seems an autonomous clock. It is the first time such a phenomenon is described in hydrozoan. Note, hydrozoan have lost key components of the animal circadian clock (the transcription regulators Clock and Cyc/Bmal1). The manuscript then takes us through a series of well-thought experiments to demonstrate that the rhythm is endogenous (it is not driven by an external rhythmic stimulus) and to some extent is self-sustained, although light-dependent. The rhythm is entrainable by a light-dark cycle (although the limits and the strength of entrainment are quite fuzzy) but it is not temperature compensated (the period of the rhythm changes extensively with temperature). Thus, we have a rhythmic process that is no longer an 'hourglass' (in C. hemispherica rhythmicity would terminate without the daily resetting by light) but it is not a fully-fledged circadian clock (self-sustained, entrainable and temperature compensated) either. Physiological experiments then show how such a clock could emerge from the hourglass mechanism. In C. hemisphaerica light triggers the release of MIH at a time when the oocytes are ready to respond to it. In C. sp. IZ-D, the oocytes acquire that competence more slowly, requiring, under light conditions (light-dark or constant light), at least 18h from the previous spawning. Thus, the addition of a delay separating two interacting components: production/release of MIH and oocytes development (ultimately resulting in their MIH-responsiveness), bend into a cycle an otherwise linear process (oocytes development).

In its essence a circadian clock is composed by an effector process (producing the output) and a modulatory process. What makes of two linear processes a cycle is the timing of their interaction. There must be a period when the two processes cannot interact (delay) and that interval must be determined by additional independent processes (cooperativity). The best-known clocks are based on transcription translation feedback loops (with such an organisation), but it is the organisation, not the actual elements, what makes the clock.

However, in C. sp. IZ-D, light is required to synchronise the development of the oocytes and temperature has a large effect on their maturation timing. Thus, additional control elements, which could contribute 'cooperativity', are not independent, explaining why such a clock does not graduate as a fully-fledged circadian clock.

In summary, the manuscript describes an interesting phenomenon that goes well beyond the immediate curiosity of hydrozoan enthusiasts. It is well written and is built on a progression of technically simple (albeit laborious) but carefully thought through experiments. Far too often we are 'dazzled into submission' by technical tour-de-force of the latest omics, parading brute 'funding' power rather than ingenuity. Here it is the opposite, and it is refreshing!

Reviewer #2: Kitsui and colleagues describe a new species of Clytia and its interesting dark spawning pattern, which contrasts with morning spawning in the more widely used Clytia hemisphaerica. I found this manuscript to be quite fascinating, carefully done and well communicated. The experiments are robust, and the quantifications seem appropriate. The schematics throughout the paper make it easier to follow. The core finding is a dual influence of light exposure on oocyte growth and the acquisition of its ability to respond to meiotic inducing hormone, resulting in a consistent spawning event 14 hours after the previous dark to light transition. These conclusions are reached by a clever series of dark/light manipulations, ovary explant and oocyte isolations, in vitro treatments with the maturation inducing hormone, and light microscopy. I have only minor suggestions and support publication but do suggest one experiment that could help with the interpretations, if it is feasible to do so.

Specifically, I am left wondering: how does the dark-light transition stall/delay MRC acquisition? What is the step that is regulated by light exposure? The authors conclude that MRC is acquired "autonomously" within oocytes by extracting oocytes at 17hps under a L12/D12 cycle. But I would be quite surprised if the oocytes themselves are detecting light (through an opsin?) and delaying their acquisition of MRC (by repression of a receptor? Something else?). Could delay of MRC instead require some signal from ovarian somatic cells to the oocyte? For the experiments in Fig. 6, oocytes received the dark to light stimulus several hours before they were removed from the gonad - time for such a signal to occur. Could the authors perform an experiment as in Figure 6, but by extracting oocytes immediately upon light exposure? Under these conditions, would the oocytes acquire MRC faster than if they remained in the ovary longer after light exposure?

Minor Point:

Page 14: References to Suppl. Fig. 1 appear to be incorrect - should they be Suppl. Fig 2?

Reviewer #3: This article was a "tour de force" in the elucidation of a clock mechanism synchronizing gamete release in a new species of hydrozoan jellyfish. I found the work to be clearly explained, comprehensive, and well executed. I have only very minor comments and essentially feel that the manuscript is ready for publication. I very much enjoyed reading this paper.

My main comment is that I am uncomfortable with the way the authors use the term "zeitgeber" to refer to a light-dark regime (p 7, and throughout). The term is more commonly used to refer to the stimulus like "light." I would prefer if the authors replace "zeitgeber" with "light-dark cycle" or "light regime" in most places. Alternatively, on p26, they refer to a "zeitgeber cycle." I haven't seen that usage before, but it seems reasonable.

It might be worth including an explicit discussion of temperature compensation somewhere in the manuscript. Some of the described processes are clearly temperature-sensitive (e.g, p 8). It would be helpful to provide the context that temperature compensation is a canonical feature of "true" circadian rhythms. Along these lines, the authors use the term "quasi circadian" (p 12), which could benefit from some definition or discussion.

I found the second paragraph of the section "MIH-induced maturation is conserved" to be a little unsatifying. [as an aside: p 19: though BMAL/CLOCK are absent, other conserved transcription factors may play a similar role (e.g., PAR-bZip family)...this might be mentioned (optional)]. The authors talk about how there may not be a novel clock mechanism, suggest that hormones may be gradually released, but never really resolved the question as to how this would create such precise timing. I think it would be ok if a sentence were added to indicate that this aspect is still unresolved/mysterious.

Other small suggestions:

p 3 "MIH comprises...." (should this be "is comprised of"?)

p 5 It's a little clear what the authors mean in the last line but "circadian clock-independent" Perhaps they mean not regulated by clock/cycle, but they have clearly demonstrated an autonomous cycle with a ~daily free-running rhythm. Sentence could use minor re-wording.

p 12 a small syntax issue in the last line. "were isolated...and tested their response..."

p 17: consider replacing "adjusted" with "entrained"

p 24...here authors mention an "autonomous circadian cycle"...but in other places, they discuss clock-independent (p 5...though maybe this is meant to be CLOCK-independent) and "quasi circadian"...if they do want to say circadian here, there is the caveat that it doesn't seem to be temperature compensated.

---

## [Editor Report · Decision Letter 2]

27 Oct 2025

Dear Tsuyo,

Thank you for your patience while we considered your revised manuscript "A light-entrained clock mechanism in a hydrozoan jellyfish that synchronises evening gamete release." for publication as a Research Article at PLOS Biology. This revised version of your manuscript has been evaluated by the PLOS Biology editors and the Academic Editor, who is fully satisfied by the changes made in response to the last round of review.

Based on our Academic Editor's assessment of your revision, we are likely to accept this manuscript for publication. However, before we can editorially accept your study we need you to address a few last data and other policy-related requests in a revision that we anticipate will not take very long. These are detailed below.

**IMPORTANT: Please address the following editorial requests.

1) TITLE: We tend to use US English in our titles, and think your title could be streamlined a bit (by removing 'that'). If you agree, we suggest you change your title to:

"A light-entrained clock mechanism in a hydrozoan jellyfish synchronizes evening gamete release"

2) ETHICS STATEMENT: We understand that your study does not need an ethics statement for the animal work, but since the jellyfish used here were collected in the wild, we ask that you provide a field license number and indicate the granting body for that ecologically sensitive field work.

3) DATA: Thank you for providing the underlying data for your study as a deposition to Zenodo. For some reason I had trouble finding your dataset when I searched the DOI. Can you provide me with a URL link?

Please do take a moment to make sure that your deposition includes the data underlying all your figures (Note that we do not require all raw data. Rather, we ask that all individual quantitative observations that underlie the data summarized in the figures and results of your paper be made available). You can read the full details of our data policy, which requires that all data be made available without restriction, here: http://journals.plos.org/plosbiology/s/data-availability. For more information, please also see this editorial: http://dx.doi.org/10.1371/journal.pbio.1001797

Please add a sentence to each figure legend pointing readers to the underlying data.

4) CODE: Per journal policy, if you have generated any custom code during the course of this investigation, please make it available without restrictions. Please ensure that the code is sufficiently well documented and reusable, and that your Data Statement in the Editorial Manager submission system accurately describes where your code can be found.

We expect to receive your revised manuscript within two weeks.

*Published Peer Review History*

*Press*

Sincerely,

Luke

Lucas Smith, Ph.D.

Senior Editor

lsmith@plos.org

PLOS Biology

---

## [Editor Report · Decision Letter 3]

31 Oct 2025

Dear Tsuyo,

Thank you for the submission of your revised Research Article "A light-entrained clock mechanism in a hydrozoan jellyfish synchronizes evening gamete release" for publication in PLOS Biology. On behalf of my colleagues and the Academic Editor, Mariana Federica Wolfner, I am pleased to say that we can in principle accept your manuscript for publication, provided you address any remaining formatting and reporting issues. These will be detailed in an email you should receive within 2-3 business days from our colleagues in the journal operations team; no action is required from you until then. Please note that we will not be able to formally accept your manuscript and schedule it for publication until you have completed any requested changes.

**IMPORTANT: I did spot one issue in this revision, which I ask that you correct as you address any production requests, to come. Specifically, I noticed the link provided in your figure legends directs readers to your bioRxiv paper, and not to the underlying dataset on Zenodo (I suspect you pasted the wrong DOI?). Please update this to include the DOI for your dataset.

PRESS

Sincerely, 

Luke

Lucas Smith, Ph.D.

Senior Editor

PLOS Biology

lsmith@plos.org